# Advances in Metal-Organic Frameworks MIL-101(Cr)

**DOI:** 10.3390/ijms23169396

**Published:** 2022-08-20

**Authors:** Minmin Zou, Ming Dong, Tian Zhao

**Affiliations:** School of Packaging and Materials Engineering, Hunan University of Technology, Zhuzhou 412007, China

**Keywords:** MIL-101(Cr), adsorption, catalysis

## Abstract

MIL-101(Cr) is one of the most well-studied chromium-based metal–organic frameworks, which consists of metal chromium ion and terephthalic acid ligand. It has an ultra-high specific surface area, large pore size, good thermal/chemical/water stability, and contains unsaturated Lewis acid sites in its structure. Due to the physicochemical properties and structural characteristics, MIL-101(Cr) has a wide range of applications in aqueous phase adsorption, gas storage and separation, and catalysis. In this review, the latest synthesis of MIL-101(Cr) and its research progress in adsorption and catalysis are reviewed.

## 1. Introduction

Metal–organic frameworks (MOFs) materials, also known as porous coordination polymers (PCPs), are an emerging and very affluent class of microporous materials [1,2,3]. In the past 30 years, scientific research reports on the topology and potential applications of MOF materials have increased almost geometrically (as shown in Figure 1). It is a group of crystal materials with a three-dimensional pore structure composed of metal atoms and organic ligands. The spatial pairing of metal atom centers and a wide variety of organic ligands give the material a controllable pore size and add many unique physicochemical properties [4,5]. An essential feature of metal–organic framework materials is their ultra-high porosity (where the free volume can be as high as 90%) and impressive Langmuir specific surface area (Langmuir specific surface area can even exceed 10,000 m^2^ g^−1^) [6,7,8]. This characteristic makes MOF materials play a crucial role in functional applications such as in the storage and separation of gases [9,10], sensing [11,12], proton conduction [13,14], and drug transport [15,16].

Materials of Institute Lavoisier Frameworks (MIL) materials are one of the most studied materials for MOFs. M(III) terephthalates (M = Cr, Fe, Al, V, Mn, and In in decreasing order of importance as well as some others) together with terephthalate derivatives and elongated terephthalate analogs form a particularly important sub-class of MOFs. The four best-known porous M(III) terephthalates (and terephthalate analogs) are MIL-47/MIL-53, MIL-88, MIL-100, and MIL-101. They are among the most recognized MOF types, especially regarding potential uses. Most of the MIL series materials use Cr^3+^, Fe^3+^, and Al^3+^ as metal ion clusters with terephthalate derivatives and terephthalate analogs as organic ligands to ligand [5]. The MIL-101 series MOFs all have similar zeolite topology but differ in surface morphology, density, and pore size. For example, MIL-101(Fe) and MIL-101(Cr) have the same topology and framework structure, and both of them are well studied. MIL-101(Fe) is composed of Fe(III) octahedral chains as secondary building units (SBU) and 1,4-benzenedicarboxylic acid [17]. MIL-101(Fe) has good catalytic properties, and under certain conditions, part of the Fe^3+^ in MIL-101(Fe) will be converted to Fe^2+^, which can play a good activation role in catalytic applications.

MIL-101(Cr) is one of the most representative materials of the MIL series and one of the most investigated MOFs today. Scientific research reports on the topology and potential applications of MIL-101(Cr) materials have continued to grow over the last 30 years (as shown in Figure 1). MIL-101(Cr) is formed by coordination of Cr_3_O ionic cluster with terephthalic acid (H_2_BDC), with the formula [Cr_3_(O)X(BDC)_3_(H_2_O)_2_] Microwave Irradiation (where BDC is terephthalic acid and X is OH^−^ or F^−^) [18], and its structure is similar to the MTN zeolite topology, as shown in Figure 2a. MIL-101(Cr) possesses two different sizes of mesoporous cage cavities with diameters of 29 Å and 34 Å (Figure 2b), and the pore windows can reach 16 Å in diameter, with a Brunauer–Emmet–Teller (BET) specific surface area of 4100 m^2^ g^−1^. MIL-101(Cr) has crystalline water molecules at the end of its molecular structure, which can be removed under high temperature or vacuum conditions, causing MIL-101(Cr) to have unsaturated metal sites (i.e., possessing potential Lewis acidic sites) [19]. MIL-101(Cr) has very high porosity, good physicochemical properties, and chemical stability; thus, it is widely used in electrocatalysis [20], photocatalysis [21], pollutant adsorption [22], mixed matrix membranes [23], detection [24], drug transport [25], and other important fields.

However, reviews on the synthesis and applications of MIL-101(Cr) are still rare. Hence, in this paper, we review the progress of research on the synthesis and application of MIL-101(Cr). The rapid development of this field, especially regarding synthetic approaches, calls for periodic updates and the development of new viewpoints. This review focuses on synthesis strategies and applications of MIL-101(Cr), especially focusing on the field of adsorption and catalysis. Additionally, the outlooks of the field, challenges of industrial preparation, and potential applications are the topics of particular interest.

## 2. Synthesis of MIL-101(Cr)

The different synthesis and activation conditions have a significant impact on the morphology, specific surface area, yield, stability of the structure, and crystallinity of MIL-101(Cr) materials. Consequently, the synthesis method is an important factor affecting the characteristics of MIL-101(Cr). At present, the main synthesis methods are hydrothermal synthesis, the solvothermal method, the microwave-assisted method, and the template method. Table 1 summarizes some advances in different synthesis methods for MIL-101 (Cr).

### 2.1. Hydrothermal Synthesis Method

#### 2.1.1. Traditional Hydrothermal Method

Hydrothermal synthesis is a common method for synthesizing nanomaterials, predominantly using water as the solvent, configuring the reaction materials into a solution, heating it to a certain temperature in a hydrothermal kettle, and standing the kettle to retain the synthesis system at a certain pressure. Utilizing the hydrothermal synthesis method, porous nanomaterials with high crystallinity and excellent properties are often obtained, which is also the most conventional method in the synthesis of MIL-101(Cr) [18]. It consists of transferring a mixed solution of Cr(NO_3_)_3_·9H_2_O, terephthalic acid (H_2_BDC), deionized water, and hydrofluoric acid (HF) into a stainless steel reaction vessel lined with polytetrafluoroethylene and heating the reaction at 220 °C for 8 h. The product is then purified using ammonium fluoride and ethanol successively, and the final product is obtained after drying in a vacuum oven (Figure 3). In this reaction system, hydrofluoric acid was used as an additive to improve the crystallinity of MIL-101(Cr) and increase the specific surface area and pore volume of the product MIL-101(Cr) during the synthesis process [18]. As hydrofluoric acid is highly toxic and volatile [44], additional protective equipment and safety precautions are essential for the synthesis of MIL-101(Cr) using hydrofluoric acid in large quantities, which undoubtedly increases the cost of the synthesis. Most importantly, most scientists have duplicated Férey’s method for the synthesis of MIL-101(Cr) without being able to obtain a high-quality product, such as the one synthesized by Férey et al. [27,31,45,46,47]. Therefore, several scientists have tried to use some additives instead of the highly toxic hydrofluoric acid to optimize the synthesis technique of MIL-101(Cr) and to upgrade the properties of MIL-101(Cr), as shown in Table 1.

Pan et al. [48] used a hydrothermal method to synthesize MIL-101(Cr) with Tetramethylammonium hydroxide (TMAOH) as an additive, as shown in Figure 4. The effect of TMAOH on the structure, morphology, and properties of MIL-101(Cr) was investigated by controlling the addition amount of TMAOH. From the SEM images, it can be seen that the quantity of TMAOH addition has a great influence on the morphology of MIL-101(Cr). With the addition of TMAOH, the morphology of the crystals changed from a smooth surface octahedral structure to a broken octahedral structure slowly, and some of the crystals dissolved into small irregular particles. The adsorption capacity of the samples on toluene was examined, and the results showed that the adsorption capacity of TMAOH-2@MIL-101(Cr) was significantly higher than that of other MIL-101(Cr) samples.

Zhao et al. [49] used sodium acetate as an additive to synthesize MIL-101(Cr) and purified it with DMF and ethanol to obtain regular octahedral MIL-101(Cr) crystals with the highest adsorption properties after activation at 140 °C and excellent cyclability. Zhao et al. [27] studied the effect of hydrofluoric acid (HF), nitric acid (HNO_3_), acidic acid (HOAc), sodium hydroxide (NaOH), and tetramethylammonium hydroxide (TMAOH) on the synthesis of MIL-101(Cr) via hydrothermal synthesis, and the analysis results revealed that NaOH as an additive can reduce the particle size of MIL-101(Cr) to nanometer size with an average particle size of 90 nm and a specific surface area of 4065 m^2^ g^−1^.

Jiang et al. [50] used monocarboxylic acid as a modulating agent to synthesize MIL-101(Cr), the BET surface area of the samples ranging from 2600 to 2900 m^2^ g^−1^ and with the particle size of 19~84 nm. The selectivity of the obtained MIL-101(Cr) towards CO_2_/N_2_ was significantly enhanced. The multihole MIL-101(Cr) was synthesized by Hu et al. [28] by using hydrochloric acid as a modulator, which had superior specific surface area and pore volume compared to HF-assisted MIL-101(Cr). The removal of hygromycin from an aqueous solution was improved by 78% for the newly synthesized sample. Frequently used additives also include acetic acid [31,47,51], nitric acid [26,52], hydrochloric acid [53,54,55], sulfuric acid [56], benzoic acid [57], sodium acetate [31,58], tetramethylammonium hydroxide [30], and phenylphosphonic acid [59]. Among them, hierarchical pore MIL-101(Cr) can also be synthesized using certain concentrations of acetic acid, tetramethylammonium hydroxide, and phenylphosphonic acid.

#### 2.1.2. Microwave-Assisted Hydrothermal Method

The microwave-assisted synthesis method refers to the synthesis of MIL-101(Cr) in a hydrothermal environment under microwave conditions by using rapid microwave heating. This method can improve the efficiency and reduce the synthesis time of MIL-101(Cr), which is mainly due to the fact that microwaves heat the solvent and improve the nucleation rate. Soltanolkottabi et al. [39] synthesized MIL-101(Cr) by the microwave-assisted method as well as electric heating method in two steps, as shown in Figure 5. This method not only significantly reduces the synthesis time of MIL-101(Cr) but also controls the morphology of the product’s crystals. The experimental results showed that increasing the pH value to 3 during the electric heating stage resulted in octahedral crystals and possessed a superior CO_2_ adsorption capacity of 7.6 mmol g^−1^ at room temperature.

Khan et al. [38] systematically compared the factors of water concentration and pH value for the impacts on the synthesis of MIL-101(Cr) via the microwave-assisted (MW) and electric heating methods. Generally, microwave-assisted MIL-101(Cr) has a smaller particle size, larger BET surface area, and pore volume compared with electrically-heated MIL-101(Cr). Meanwhile, a high water concentration and pH value prefer the smaller particle size of MIL-101(Cr).

Zhao et al. [35] synthesized MIL-101(Cr) at 220 °C using 300 W microwave irradiation for 60 min. The sample had an octahedral morphology with a particle size of 100 nm, possessing a BET specific surface area of 3054 m^2^ g^−1^ and a pore volume of 2.01 cm^3^ g^−1^. The synthesized MIL-101(Cr) demonstrated a high benzene adsorption capacity of 16.5 mmol g^−1^. Yin et al. [36] successfully synthesized MIL-101(Cr) with standardized and homogeneous crystals by using the microwave-assisted method with a reaction time of only 30 min at 220 °C via 400 W microwave radiation, which greatly reduced the synthesis time of MIL-101(Cr).

#### 2.1.3. Template Hydrothermal Method

The template method is an important method for controlling the morphology and dimensions of crystals, and it is classified into hard and soft templates depending on the characteristics of the template itself and its domain-limiting ability [60]. In general, the template method enables the synthesis of hierarchically porous MIL-101(Cr) (HP-MIL-101(Cr)), which improves the properties of MIL-101(Cr) materials and expands their applications [40]. Using a template to occupy space in the crystal framework of MIL-101(Cr), removal of the template leads to extra mesoporous or macroporous structures [61].

Yang et al. [40] prepared nanoscale MIL-101(Cr) crystals containing macroporous structures using expanded graphite (EG) as a template. The synthesized MIL-101(Cr) has a BET specific surface area of 3751 m^2^ g^−1^ with a yield of 43%. More importantly, the reaction time was only 2 h, which was only one-fourth of the conventional method (8 h). Huang et al. [43] synthesized MIL-101(Cr) with a hierarchical porous structure by using cetyltrimethylammonium bromide (CTAB) as a soft template. Unlike the conventional micron-scale ortho-octahedral morphology, MIL-101(Cr), with the addition of CTAB, showed irregular nanoparticles (Figure 6). HP-MIL-101(Cr) has a wide distribution of mesoporous and macroporous structures with a microporous-to-mesoporous ratio of 19:1.

### 2.2. Solvothermal Method

The solvothermal method involves the reaction of metal salts, organic ligands, and solvents (non-aqueous or organic) in a certain ratio to produce MOF crystals. This method is usually performed at higher temperatures or under steam conditions to make the reaction occur through intermolecular contact. The presence of high temperatures and pressures allows for a much higher solubility of the metal salt in the solvent and a faster reaction rate.

High yields of MIL-101(Cr) can be obtained at lower temperatures using the solvothermal method. Tan et al. [34] investigated the synthesis of MIL-101(Cr) by a mixed-solvent thermal method using Cr(NO_3_)_3_·9H_2_O, terephthalic acid (H_2_BDC), hydrofluoric acid (HF) as raw materials, and DMF and H_2_O in different volume ratios as mixed solvents. The effect of temperature on the synthesis of MIL-101(Cr) was investigated, and the results showed that MIL-101(Cr) could be synthesized at a low temperature of 140 °C using the mixed-solvent thermal method. Furthermore, the volume ratio of DMF and H_2_O is an important factor affecting the formation of MIL-101(Cr). At DMF/H_2_O = 0.20, the product possessed a BET specific surface area of 2453 m^2^ g^−1^, and the yield was as high as 83.3%. The static adsorption results showed that the capacity of water absorption of MIL-101(Cr) synthesized using a mixed-solvent thermal method was also higher than that of the MIL-101(Cr) synthesized by the conventional method. Fallah et al. [62] synthesized MIL-101(Cr) and the composite MOR/MIL-101(Cr) with filamentous zeolite (Mordenite Zeolite, MOR) by the solvothermal method, and the thermogravimetric analysis test showed that MOR/MIL-101(Cr) is statistically more stable than MIL-101(Cr).

At present, the hydrothermal method is still the most commonly used strategy for MIL-101(Cr) synthesis, which possesses a high specific surface area and good crystallinity. Furthermore, numerous scientists have attempted to use other additives to replace the originally reported HF during MIL-101(Cr) synthesis. For example, the use of HNO_3_ could increase the yield by over 80%, the use of acetic acid could achieve a nano-sized product, etc. The microwave-assisted method would largely accelerate the reaction process and save time, while the template method could provide special structural MIL-101(Cr) crystals, and the solvothermal method could decrease the reaction temperature to as low as 140 °C. Thus, the researchers could choose the appropriate synthesis method according to their needs and conditions.

## 3. Applications

MIL-101(Cr) has an ultra-high specific surface area and good hydrothermal/water stability, thus, demonstrating a wide range of applications in adsorption and catalysis. This review mainly focuses on adsorption, catalysis, and other applications of MIL-101(Cr), which is shown in Figure 7.

### 3.1. Adsorption

Due to the characteristics of high specific surface area (4100 m^2^ g^−1^), excellent stability, and a large number of unsaturated metal sites, MIL-101(Cr) is quite suitable for adsorption of gases, dyes, or water vapor [48,63,64].

#### 3.1.1. Gas Adsorption

H_2_ is an ideal, safe, and green energy source, but it is extremely unstable and difficult to store and transport under normal temperature and pressure. Under normal conditions, the storage and transport of H_2_ can be achieved by using adsorbents to adsorb H_2_ [65]. CO_2_ is a greenhouse gas and a major source of the greenhouse effect, so capturing CO_2_ is an inevitable trend [66]. Metal–organic framework materials with large specific surface area and a high void fraction have great potential in the field of adsorption and storage of gases. Hong et al. [67] tested and compared the CO_2_ capture capacity of MIL-101(Cr) with that of 13× zeolite monomer. The results showed that the adsorption capacity of MIL-101(Cr) for CO_2_ was 37% higher than that of a 13× zeolite monomer, and the adsorption efficiency was 1.5 times higher than that of the 13× zeolite monomer. Moreover, compared with other metal–organic frameworks, MIL-101(Cr) has a better adsorption capability for gases, such as CO_2_, H_2_, and CH_4_ [33,68,69].

Yang et al. [68] investigated the adsorption performance of MIL-101(Cr), ZIF-8, and UiO-66 on N_2_, CH_4_, and CO_2_. It was shown that MIL-101(Cr) had the best adsorption performance for the above three gases, especially for CO_2_, where the adsorption of MIL-101(Cr) (29.4 mmol g^−1^) was 2.19 and 3.13 times higher than that of UiO-66 (13.4 mmol g^−1^) and ZIF-8 (9.4 mmol g^−1^), respectively (Figure 8).

Llewellyn et al. [33] comparatively studied the adsorption performance of MIL-101(Cr) and MIL-100(Cr) on CO_2_ and CH_4_. The results showed that the adsorption performance of MIL-101(Cr) was much higher than that of MIL-100(Cr), and its maximum adsorption capacity for CO_2_ could reach 40 mmol g^−1^, which was over twice that of MIL-100(Cr) (18 mmol g^−1^). A similar trend was also detected in the case of H_2_ adsorption; MIL-101(Cr) showed a higher adsorption capacity than that of MIL-100(Cr) [69].

MIL-101(Cr) has been studied extensively in the gas adsorption field, and Table 2 shows the various gas adsorption capacities of MIL-101(Cr) in recent years. As early as 2009, the gas adsorption capacity of MIL-101(Cr) was investigated by Chowdhury et al. [70]. They measured the adsorption characteristics of MIL-101(Cr) on CO_2_, CH_4_, C_3_H_8_, SF_6_, and Ar gases at different temperatures by using the weight method. The results showed that MIL-101(Cr) had good adsorption performance for all five gases, and the best adsorption performance was achieved at 283 K. Munusamy et al. [71] investigated the adsorption characteristics of MIL-101(Cr) with different forms for CO_2_, CO, CH_4_, and N_2_ at different temperatures. For the four gas molecules, MIL-101(Cr) revealed very good adsorption properties, and the adsorption capacity tended to decrease with the increasing temperature. Among the four gases, MIL-101(Cr) showed the best adsorption toward CO_2_, which was three times or even six times higher than other gases. Meanwhile, the experiments showed that the adsorption capability of MIL-101(Cr) with the powder type was higher than those of MIL-101(Cr) with the particle type. Montazerolghaem et al. [72] also investigated the effect of different forms of MIL-101 (Cr) on its adsorption properties. The results presented that the powder form of MIL-101(Cr) has higher adsorption performance than the granular form of MIL-101(Cr), and the adsorption of CO_2_ at 7.1 bar and 298.2 K reached 9.72 mmol g^−1^ for the powder form, which was 1.53 times higher than that of granular MIL-101(Cr) (6.34 mmol g^−1^). The factors affecting the adsorption performance of MIL-101(Cr) include not only the temperature, pressure, and sample forms but also the MIL-101(Cr) crystals structure and the synthesis method. Chong et al. [73] proposed a solvent-free method to synthesize MIL-101(Cr). It was found that the best adsorption performance of MIL-101(Cr) was obtained at a Cr:BDC molar ratio of 1:1, with a BET specific surface area of 1110 m^2^ g^−1^ and an adsorption capacity of 18.8 mmol g^−1^ for CO_2_.

The adsorption capacity of MIL-101(Cr) was usually improved via chemical modification, such as by using amine functionalization [79,85,91], carbon doping [80,82,92], and metal doping [93,94]. Darunte et al. [91] investigated the adsorption capacity of amine-functionalized MIL-101(Cr) and conventional MIL-101(Cr) toward CO_2_. The results disclosed that the amine-functionalized MIL-101(Cr) possessed a higher CO_2_ adsorption capacity compared with conventional MIL-101(Cr). Zhou et al. [93] successfully synthesized magnesium-doped bimetallic MIL-101 (Cr, Mg) by adding magnesium salts in the synthesis process. The effect of the amount of doped Mg^2+^ on the adsorption properties of MIL-101(Cr, Mg) are displayed in Figure 9, indicating that the doped Mg^2+^ largely improved the adsorption capacity of MIL-101. Under the same condition, MIL-101(Cr, Mg) presented an uptake of 3.28 mmol g^−1^ of CO_2_, which was 40% higher than MIL-101(Cr). Additionally, MIL-101(Cr, Mg) also showed significantly higher selectivity for CO_2_/N_2_ compared with MIL-101(Cr).

In addition to metal doping, carbon doping was also an important method to enhance the adsorption properties of the MOFs. Zhou et al. [82] synthesized a novel composite (GrO@MIL-101(Cr)) by using graphene oxide (GrO) and MIL-101 (Cr). It was found that the adsorption capacity of GrO@MIL-101(Cr) for CO_2_ was significantly higher than that of MIL-101(Cr). Han et al. [89] used ionic liquids combined with MIL-101(Cr) to enhance the adsorption capacity of MIL-101(Cr) on NH_3_. At 298 K and 1 bar, the adsorption of NH_3_ by the composite was 24.12 mmol g^−1^, which was 2.7 times higher than that of MIL-101(Cr) (8.92 mmol g^−1^). Moreover, the composite revealed better stability and remained stable under wet NH_3_ conditions; its excellent adsorption performance in vapor atmosphere near saturated ammonia solution (23.55 mmol g^−1^) was reduced by only 2%.

#### 3.1.2. Dye Adsorption

Dyes are widely used in daily applications such as the food industry, packaging, printing, leather industry, etc. Due to the incomplete treatment of industrial wastewater, about 10–15% of the dyes consumed therein are directly discharged into the aqueous environment every year [95,96]. The discharge of large amounts of organic dyes can be extremely harmful to the environment and ecosystem [97,98]. The adsorption method has the advantages of having a simple process, high operability, and no secondary pollution and is a very effective method for dye wastewater treatment [98,99]. MIL-101(Cr) is considered to be an excellent adsorbent in dye adsorption applications due to its outstanding water/chemical stability, high porosity, and its large specific surface area [100,101]. Table 3 summarizes the studies of MIL-101(Cr) or MIL-101(Cr)-based materials for dyes adsorption.

Haque et al. [99] compared the removal performance of methyl orange (MO) from aqueous solutions by using MIL-101(Cr), functionalized MIL-101(Cr), and MIL-53(Cr). The analyzed results indicated that the functionalized MIL-101(Cr) had the highest removal ability for MO among the three MOFs. Zhang et al. [104] reported that the charge and size of MIL-101(Cr) could greatly affect its adsorption capability on different organic dyes such as methylene blue (MB), congo red (CR), and methyl orange (MO). Mahmoodi et al. [105] used a DMF-free method to synthesize MIL-101(Cr) and investigated the adsorption capacity of MIL-101(Cr) on direct red (DR80) and acid blue (AB92). It was found that MIL-101(Cr) exhibited a good adsorption capacity and cyclic adsorption for both dye solutions, with the maximum adsorption capacity of 227 mg g^−1^ for DR80 and 185 mg g^−1^ for AB92.

Shen et al. [42] reported that methyl orange (MO) and methylene blue (MB) could be efficiently removed from aqueous solutions by using MIL-101(Cr), which were synthesized with different mineralizing agents. It is worthy to note that, for the removal of MB, MIL-101(Cr) containing mesopores had a higher adsorption capacity for dyes than that of the conventional microporous MIL-101(Cr) and the higher the mesoporous ratio, the better the adsorption performance. However, the adsorption of MO was mainly dependent on the electrostatic interaction between the dye and MIL-101(Cr), not the porous structure. Huang et al. [43] also found that for the adsorption of MB in an aqueous solution, hierarchically porous MIL-101(Cr) presented with a much higher adsorption capacity than that of the conventional microporous MIL-101(Cr). Moreover, the crystal morphology of MIL-101(Cr) also affected the adsorption performance of MIL-101(Cr). Xu et al. [107] prepared MIL-101(Cr) crystals with different morphologies by varying the reaction temperature and cooling rate and produced spherical MIL-101(Cr) at 150 °C and octahedral MIL-101(Cr) at other temperatures, as shown in Figure 10a. Figure 10b showed the adsorption performance of MIL-101(Cr) samples with different temperatures for the dyes (MO and MB). It can be seen from the figure that the spherical MIL-101(Cr) had the best adsorption capacity among all of the samples, with a maximum adsorption amount of 420.2 mg g^−1^ for MO. Similarly, Zhao et al. [108] prepared spherical MIL-101(Cr) at 160 °C without any additives, and it was also found that the adsorption capacity of the spherical MIL-101(Cr) for methyl orange and rhodamine was much higher than that of the conventional octahedral MIL-101(Cr).

The introduction of functional groups (e.g., –SO_3_H [109,112], –COOH [111], –NH_2_ [113,115], etc.) into MIL-101(Cr) was a common method to improve the adsorption capacity of MIL-101(Cr). Yang et al. [112] reported an MIL-101(Cr)-SO_3_H material, which showed excellent adsorption performance for organic dyes in an aqueous solution. The -SO_3_H group increased the electrostatic interaction and hydrogen bonding between the adsorbent and linear anionic dyes.

Yang et al. [111] conducted adsorption experiments using MIL-101(Cr)-COOH on three dyes, including Congo Red, Methyl Orange, and Acid Chromium Blue K. Compared with conventional MIL-101(Cr), the adsorption capacity of MIL-101(Cr)-COOH on Congo Red and Methyl Orange was significantly higher, while for the adsorption of Acid Chrome Blue K, MIL-101(Cr)-COOH displayed a decreased adsorption performance. That was because the –COOH group increased the electrostatic and hydrogen bonding forces between MIL-101(Cr) and linear anionic dyes, which improved the adsorption capacity toward the linear anionic dyes. Meanwhile, it also increased the spatial site resistance effect between MIL-101(Cr) and nonlinear anionic dyes, which caused a decrease in the adsorption of nonlinear anionic dyes. Zhang et al. [113] found that the –NH_2_ group could increase the adsorption capacity of the MIL-101(Cr)-NH_2_ for linear anionic dyes by modulating the driving forces (electrostatic, hydrogen π–π stacking interactions, pore volume, and spatial site resistance) between the adsorbent and the dyes, and its adsorption capacity on congo red and methyl orange was increased compared to the pristine MIL-101(Cr) by 1.17 and 1.02 times. Yang et al. [112] used sulfonyl modification of MIL-101 to produce MIL-101-SO_3_H, which was found to have excellent adsorption properties for organic dyes. The experimental adsorption capacities of MIL-101-SO_3_H for methyl orange, congo red, and acid chromium blue K were 688.9, 2592.7, and 213.2 mg g^−1^, which were 69.6%, 89.6%, and 51.5% higher than those of unmodified MIL-101, respectively. As can be seen in Table 3, the adsorbent had the best dye adsorption performance in MIL-101(Cr) and derivatives. The uncoordinated –SO_3_H group increased the electrostatic attraction and hydrogen bonding between MIL-101-SO_3_H adsorbent and linear anionic dyes, thus increasing the adsorption capacity of the linear anionic dyes.

MIL-101(Cr)-based composites usually disclosed significantly higher adsorption performance compared with pure MIL-101(Cr). For instance, Vo et al. [106] prepared a series of GrO/MIL-101(Cr) composites (GrO = graphite oxide), which were applied to the removal of contaminants such as methyl orange and reactive blue 198 (RB198) (Figure 11). It was found that the 6 wt% GrO-loaded GrO/MIL-101(Cr) composites had the best adsorption capacity for the dyes, with the adsorption amounts of 235 mg g^−1^ and 175 mg g^−1^ for MO and RB198, respectively, which were 2.3 and 1.97 times higher than that of the pure MIL-101(Cr). Wu et al. [117] reported a TiO_2_/MIL-101(Cr) composite, which demonstrated good adsorption performance for MO, and the adsorption capability could reach 242.02 mg g^−1^ in 70 mg L^−1^ MO solution.

#### 3.1.3. Drug Adsorption

In addition to the dyes, MIL-101 (Cr) or MIL-101(Cr)-based materials also performed well for the removal of drugs (e.g., antibiotic drugs and pesticide residues). Table 4 lists the summary of the application of MIL-101(Cr) or MIL-101(Cr)-based materials in the adsorption and removal of drugs.

Hu et al. [28] studied the adsorptive removal of oxytetracycline (OTC) from an aqueous solution by MIL-101(Cr) synthesized with different mineralizers. Moreover, the results showed that MIL-101(Cr) synthesized with HCl as a mineralizer possessed a higher adsorption capacity than that of MIL-101(Cr) synthesized with HF as a mineralizer. Huang et al. [118] used MIL-101(Cr) for the adsorptive removal of sulfamethoxazole (SMZ) from water and discovered that the adsorption of SMZ by MIL-101(Cr) was spontaneous and exothermic with a fast adsorption rate, reaching saturation adsorption within 180 s. The largest adsorption capacity of MIL-101(Cr) for SMZ was 181.82 mg g^−1^. Moreover, it was also found that MIL-101(Cr) also had a good adsorption capacity for antibiotic drugs such as sulfamonomethoxine (SCP), sulfamonomethoxine (SMM), and sulfadimethoxine (SDM). Shadmehr [119] reported the adsorptive removal of propiconazole fungicides from an aqueous environment using MIL-101(Cr), and the maximum adsorption amount of propiconazole could reach 89.78 mg g^−1^. Mirsoleimani-azizi et al. [120] employed MIL-101(Cr) for the removal of diazines from aqueous solutions and found that the removal of diazines could reach 92.5%, which indicated that MIL-101(Cr) revealed promising application for agricultural wastewater treatment.

Isiyaka et al. [121] introduced MIL-101(Cr) as an adsorbent for the effective removal of 4-chloro-2-methylphenoxyacetic acid (MCPA) from an aqueous solution. The rapid removal of MCPA by MIL-101(Cr) was recorded within 25 min, and the maximum adsorption capacity of MIL-101(Cr) for MCPA was 233.576 mg g^−1^; the removal rates could be over 90%. Li et al. [125] investigated the adsorptive removal of ciprofloxacin (CIP) from water by MIL-101(Cr)-HSO_3_, and the possible adsorption mechanism of CIP on MIL101(Cr)-HSO_3_ are shown in Figure 12. The results showed that MIL-101(Cr)-HSO_3_ had a good adsorption capacity for CIP with a maximum value of 564.9 mg g^−1^, which was considered to be one of the best materials for CIP removal.

In order to improve the adsorption capacity of MIL-101(Cr), Jin et al. [128] used MIL-101(Cr) loaded with Cu/Co bimetallic particles to produce a new Cu@Co/MIL-101(Cr) composite. The adsorption of tetracycline (TC) of Cu@Co/MIL-101(Cr) was much higher than that of pure MIL-101(Cr), and the maximum adsorption capacity of the composite could reach 225.179 mg g^−1^. Cu@Co/MIL-101(Cr) had a stronger adsorption performance due to the change of electronegativity and the enhanced electrostatic interaction with TC after doping with Cu/Co metal particles (Figure 13).

#### 3.1.4. Other Adsorption Applications

Volatile organic compounds (VOCs) are a major source of air pollution, which not only aggravate ozone layer depletion and the greenhouse effect but also endanger human health [133,134]. MIL-101(Cr) or MIL-101(Cr)-based materials were also employed in the field of VOC removal. Bullot et al. [135] investigated the adsorption capacity of MIL-101(Cr) on (poly)chlorobenzene pollutants. It was found that MIL-101(Cr) had an excellent adsorption capability toward chlorobenzene pollutants due to the extremely strong π–π interactions between the MIL-101(Cr) and the chlorobenzene pollutants. Furthermore, the adsorption capacity of nano-sized MIL-101(Cr) for both 1,2-dichlorobenzene and 1,2,4-trichlorobenzene was significantly higher than that of micron-sized MIL-101(Cr). Shafiei et al. [136] investigated the adsorption capacity of MIL-101(Cr) for different gaseous VOCs. The results revealed that MIL-101(Cr) exhibited an excellent adsorption capacity toward all of the studied VOCs, especially the absorption rate for gasoline, which could be up to 90.14 wt%, which was 3.6 times higher than that of commercially available activated carbon. Heydari et al. [137] investigated the adsorptive removal of toluene from aqueous solutions by MIL-101(Cr) via response surface methodology. It was found that the removal of toluene from an aqueous solution by MIL-101(Cr) could reach 97% under the selected condition.

MIL-101(Cr) can also be applied to remove heavy metal ion contamination from water bodies. Josep and his colleagues [138] analytically investigated the adsorption capacity of MIL-101(Cr) on Cu^2+^, Cd^2+^, and Pb^2+^ in aqueous solutions. The maximum adsorption amounts of Cu^2+^, Cd^2+^, and Pb^2+^ were 16,099 mg g^−1^, 15,769 mg g^−1^, and 19,043 mg g^−1^, respectively, which were higher than most adsorbents on the market, especially the adsorption amount of Cu^2+^ could reach 100 times of the adsorption amount of prulan/polydopamine hydrogel (100.9 mg g^−1^) [139]. It revealed that the adsorption of heavy metals from an aqueous solution using MIL-101(Cr) was mainly attributed to the electrostatic interaction between them, indicating that MIL-101(Cr) could effectively remove heavy metals from an aqueous solution. The metal removal ability of MIL-101(Cr) can be enhanced by functionalization. Rastkari et al. [140] reported that the adsorption and removal ability of tetraethylenepentamine (TEPA)-grafted MIL-101(Cr) on metals in an aqueous solution was investigated. It was shown that the grafted TEPA-MIL-101(Cr) had an excellent adsorption capacity for Pb^2+^, Cu^2+^, Cd^2+^, and Co^2+^ in an aqueous solution, and the adsorption capacity exceeded that of the original MIL-101(Cr) by a factor of eight. When applied to real water, TEPA-MIL-101(Cr) can remove more than 95% of the metals in the water.

### 3.2. Catalysis

MIL-101(Cr) and MIL-101(Cr)-based materials can be used as a catalyst in many reactions due to their high porosity and the potential unsaturated metal sites in the structure (Table 5). In this section, the applications of MIL-101(Cr) and MIL-101(Cr)-based materials in various reactions such as oxidation, condensation, C-C coupling, hydrogenation, acid-base synergy, ring opening, etc. will be reviewed.

#### 3.2.1. Oxidation of Olefins and Aromatic Heterocycles

Maksimchuk [153] prepared a hybrid material, PWx/MIL-101(Cr), containing 5~14 wt% of tungstate oxide (PWx), which was used as the catalyst in the oxidation of cycloethylene to epoxycyclohexane. It was found that PWx/MIL-101(Cr) exhibited very good catalytic activity in the reaction, even close to the pure PWx. Additionally, PWx/MIL-101 showed quite good catalytic activity in the epoxidation of various olefins, including 1-octene, cyclooctene, limonene, etc. Furthermore, in the oxidation of substrates with aromatic groups such as styrene, PWx/MIL-101(Cr) also displayed high catalytic efficiency. Leng [45] investigated the catalytic activity of MIL-101(Cr) in the oxidation of cyclic ethylene, which disclosed excellent catalytic activity with a TOF value of 1.70 h^−1^. Interestingly, it was reported that the nano-sized MIL-101(Cr) presented higher catalytic activity than that of micro-sized MIL-101(Cr) in the oxidation of 1-dodecene [100,154].

Yeganeh [141] reported the effect of MIL-101(Cr) and MIL-100(Fe) and their composites with copper phthalocyanine (CuPc) as catalysts in the styrene oxidation reaction (Figure 1). The styrene conversion was significantly improved by the addition of an activated MIL-101(Cr) catalyst, especially in the presence of CuPc@MIL-101(Cr); the conversion of styrene could reach 100% with a selectivity of 85% for styrene oxides. Mortazavi et al. [155] found that MIL-101(Cr)-SO_3_H was an efficient catalyst in the oxidative styrene cleavage, the conversion of the reactant reached 99% in 20 min and the selectivity of 2-methoxy-2-phenylethanol was 100% (Figure 2). Santiago-Portillo and coworkers [151] investigated the catalytic property of MIL-101(Cr)-X (X = H, NO_2_, SO_3_H, Cl, CH_3_, and NH_2_) in the oxidation of benzylamine to the corresponding n-benzylbenzylamine. MIL-101(Cr)-NO_2_ exhibited the highest catalytic activity among the above materials, with a catalytic activity about six times higher than that of the parent MIL-101(Cr). It was disclosed that the introduction of suitable radicals on the terephthalic acid linker could modulate the electron density around Cr^3+^ and enhance the catalytic activity of MIL-101(Cr). MIL-101(Cr)-NO_2_ can also be used as a catalyst in the oxidation of thiophene or used as a radical initiator for the oxidative desulfurization of dibenzothiophene (Figure 3) [156]. Ying et al. [154] proposed a hydrophobic mesoporous silica-encapsulated MIL-101(Cr) composite, which demonstrated better catalytic activity in indene oxidation compared with pristine MIL-101(Cr). Zhao and the coworkers [59] prepared hierarchically porous (HP) MIL-101(Cr) by using phenylphosphonic acid as a modulating agent, which also presented high catalytic efficiency toward indene oxidation. That was mainly contributed to the fact of hierarchically porous structure in MIL-101(Cr) that exposed more active sites and thus exhibited better catalytic activity. Consequently, Zhao et al. [100] found that the addition of acetic acid in MIL-101(Cr) synthesis could also cause a hierarchically porous structure, which exhibited quite good catalytic activity during the oxidation of indene and 1-dodecene.

#### 3.2.2. Esterification and Acylation Reactions

Functionalized MIL-101(Cr) had good catalytic activity in the esterification and acylation reactions. Zang and colleagues [146] reported the catalytic activity of MIL101(Cr)-SO_3_H applied to the esterification reactions of alcohols and acids. It was found that pristine MIL-101(Cr) had no catalytic activity in the esterification process; however, MIL101(Cr)-SO_3_H showed pretty good catalytic activity in the esterification reaction. Meanwhile, MIL101(Cr)-SO_3_H was employed as an efficient catalyst in the esterification of cyclic ethylene with formic acid (Figure 4), which presented a high selectivity of 97.61% [157]. Khder et al. [158] fabricated MIL-101(Cr) that was loaded with 12-phosphotungstic acid (H_3_PW_12_O_40_) to work as a catalyst in the Pechmann reaction, esterification reaction, and Friedel–Crafts acylation reaction. The analyzed results demonstrated that the obtained composite revealed good catalytic activity in all of the above three reactions.

#### 3.2.3. CO_2_ Cycloaddition Reaction

Jiang and the coworkers [159] utilized cationic ionic liquids (1,1′-(*n*-hexane-1,6-diyl)-bis(3methylimidazolium) dibromide) to synthesize ionic liquid@MIL-101(Cr) composites. The ionic liquid@MIL-101(Cr) materials exhibited good catalytic properties for the CO_2_ cycloaddition reaction without any additives and solvents (Figure 5), and the product yield could reach 92.5%.

Bahadori and colleagues [160] synthesized an MIL-101(Cr) composite with carboxylic acid-based and imidazole-based ionic liquids (TSIL). Furthermore, MIL-101(Cr)-TSIL can be used as a catalyst in the reaction of CO_2_ gas with epoxy compounds without solvents. The conversion of CO_2_ could reach 95% with a selectivity of 98% within 6 h at 110 °C.

#### 3.2.4. Acetal and Condensation Reactions

Bromberg et al. [147] reported a PTA@MIL-101 composite that combined with MIL-101(Cr) and phosphotungstic acid (PTA), which showed outstanding catalytic activity for aldehyde–alcohol reactions. At the same time, a PTA/MIL-101 composite was also employed as a catalysts in the Bayer condensation reaction of benzaldehyde with 2-naphthol and the three-component condensation reaction of benzaldehyde, 2-naphthol and acetamide (Figure 6) [161]. Mortazavi and colleagues [56] investigated the catalytic property of MIL-101(Cr)-SO_3_H in the acetalization reaction of benzaldehyde with ethylene glycol (Figure 7). The conversion of benzaldehyde catalyzed by MIL-101(Cr)-SO_3_H could reach 90%, which was much higher than that of MIL-101(Cr). MIL-101(Cr)-SO_3_H also showed pretty good catalytic activity for the acetalization of benzaldehyde with ethylene glycol, with 91% of yield in 1 h at room temperature [41]. Zhao et al. [162] synthesized chitosan-coated MIL-101(Cr) nanoparticles, which exhibited excellent catalytic activity with the yield of 99% during a one-pot tandem deacetylation-knoevenagel condensation reaction (benzaldehyde dimethyl acetal releases methanol to produce benzaldehyde, and benzaldehyde undergoes synergistic dehydration with malononitrile to produce end product 2-benzylmethanecarbonitrile, Figure 7). The catalyst could be reused several times, and there was no significant catalytic activity loss after five cycles, and the catalyst was structurally stable and undamaged during the reaction, as confirmed by XRD.

#### 3.2.5. Coupling Reaction

Recently, Chen et al. [163] reported the catalytic property of MIL-101(Cr)-SO_3_H for the aerobic cross-dehydrogenation coupling (CDC) reaction (Figure 8). MIL-101(Cr)-SO_3_H exhibited quite good catalytic activity in CDC reaction with a product yield of 63% and high selectivity of 98%, which was much higher than that of typical commercial solid acid catalysts. Li et al. [143] synthesized amino-functionalized MIL-101(Cr) via reducing MIL-101(Cr)-NO_2_, which exhibited excellent catalytic performance in the Henry reaction of benzaldehyde.

#### 3.2.6. Cyanosilylation and Hydroxyalkylation Reaction

Zhang et al. [142] investigated the catalytic property of four types of MOFs, MIL-101(Cr), MIL-53(Al), MIL-47(V), and UiO-66(Zr), as catalysts in the cyanosilylation reaction of aldehydes with trimethylsilyl cyanide (TMSCN). Among the four MOFs, MIL-101(Cr) revealed the best catalytic activity with a conversion of 96%, which was due to the extremely strong interaction between the metallic chromium center of MIL-101(Cr) and the carbonyl oxygen atom of benzaldehyde. Henschel et al. [164] also found that MIL-101(Cr) exhibited excellent catalytic activity in the cyanosilylation reaction, and the benzaldehyde conversion could be up to 98.5%.

Xia and coworkers [144] reported that the Al metal-doped MIL-101(Cr/Al) displayed good catalytic activity for the hydroxyalkylation reaction of phenol with formaldehyde (Figure 9). MIL-101 (Cr/Al) had a large specific surface area and pore volume, which facilitates the process of substrate and product expulsion. The well-interconnected nanopores exposed the high density of active sites; thus MIL-101(Cr/Al) presented a high catalytic performance for the hydroxyalkylation reaction of phenol with formaldehyde.

### 3.3. Other Applications

#### 3.3.1. Drug Delivery

Traditional drug treatments were ineffective, requiring high doses, and having to be used very frequently. The use of other substances as drug carriers has been found to significantly improve the therapeutic effect of drugs [165,166]. For example, nanoparticles [167], nanofibers [168], hydrogels, and other substances can effectively wrap and release drugs, thus improving their therapeutic effect. MOFs are structurally-tunable porous materials with an adjustable pore size and a high specific surface area, hence, they have great potential in the application of drug delivery. MIL-101(Cr) was also considered to be a candidate for drug delivery. For instance, Gordon et al. [169] used MIL-101(Cr) as a carrier for the delivery of acetaminophen, progesterone, and stavudine FV. The results indicated that the loaded drugs would be slowly released in 30 min, which suggested that MIL-101(Cr) had great potential in drug delivery applications. Ayvaz Koroglu et al. [170] found that MIL-101(Cr) has a high storage capacity of 1000 mg L^−1^ for corticosteroids (i.e., desoximetasone, clobetasol propionate, methylprednisolone, and trenbolone, hydrocortisone valerate) with the controlled release of drug molecules. Horcajada’s [171] study compared the adsorption and release of MIL-101(Cr) on ibuprofen. MIL- 101(Cr) exhibited excellent drug loading and controlled release, with higher drug doses and a longer delivery time for ibuprofen. Silva et al. [172] reported that MIL-101(Cr) and MIL-101(Cr)-NH_2_ presented good loading and release capacity toward ibuprofen (IBU) and nimesulide (NMS), respectively. Although MIL-101(Cr) showed good potential for drug delivery application, however, as far as we know, it cannot be eventually commercially used for clinics yet, due to the presence of Cr in the framework.

#### 3.3.2. Sensors

The sensors mainly involved electrochemical sensors [173,174], biosensors [175], electrochemical biosensors [176,177], immunosensors [178], fluorescent sensors [179,180], etc. Sensors have penetrated into a wide range of fields, such as industrial production [181,182,183], environmental protection [184], bioengineering [179,185], medical diagnosis [186,187,188,189], marine exploration, and so on. MIL-101(Cr) also had great application value in sensing due to its characteristics [190,191]. Haghighi et al. [192] reported a new quartz gas sensor by using MIL-101(Cr) as a sensing material for the detection of formaldehyde gas in the environment. When MIL-101 (Cr) adsorbed formaldehyde molecules, the mass of the quartz crystal surface changed, and its frequency also changed, so formaldehyde gas can be detected by observing the frequency change in the sensor response. The sensor had a minimum detection limit of 1.79 ppm and had good repeatability and stability in the detection range.

Zhang et al. [193] prepared an immunosensor based on nanoparticle-loaded MIL-101(Cr) for the detection of microcystin lr in water. The sensor had good stability and practicality and had an ultra-high recovery for the detection of microcystin lr in water bodies. The MIL-101(Cr) sensor had a very good recovery with a detection recovery of 102%, which was higher than that of the nanobiosensor prepared from NiO-rGO/MXene complex (89–101%) [194]. Yang et al. [195] prepared a composite fluorescent sensor by combining amino-functionalized carbon quantum dots with MIL-101(Cr)-SO_3_H. In this system, MIL-101(Cr)-SO_3_H wrapped the amino–carbon quantum dots through the hydrogen bond between SO_3_H and NO_2_ groups, acting as a selective adsorbent to capture the target analyte. The sensor exhibited good selectivity and sensitivity for 2,4-dinitrophenol with a detection limit of 0.041 μM.

#### 3.3.3. Proton Conduction

Devautour-Vinot et al. [196] investigated the electronic conductivity of MIL-101(Cr)-NO_2_ and its propyl sulfonic acid-modified material. The conductivity of the material reached 4.8 × 10^−3^ S cm^−1^ at a temperature of 363 K and relative humidity of 95%. The proton conduction properties of the material can be further improved by impregnating it with a strong acid (H_2_SO_4_), and the conductivity could be up to 1.3 × 10^−1^ S cm^−1^. Recently, Sun et al. [197] reported a new material consisting of MIL-101(Cr) and phosphotungstic acid (HPW) with an amino acid-base adduct (HPW-SA), which showed high-temperature proton-conductivity. Since the solid–liquid phase transition of HPW-SA accelerated the motion of protons, the electron conductivity of the material increased sharply near the phase transition temperature. At 150 °C, the conductivity of the material was 3.1 × 10^−5^ S cm^−1^, while the temperature increased to 190 °C, the proton conductivity of HPW-SA@MIL-101(Cr) reached 1.1 × 10^−3^ S cm^−1^, which was higher than most of the reported high-temperature proton-conducting materials. Meanwhile, the proton conductivity of HPW-SA@MIL-101(Cr) remained stable after several cycle tests without any significant decrease, indicating excellent stability and cyclability.

#### 3.3.4. Hybrid Matrix Membranes

The MOF materials were synthesized in functional form due to their inherent porosity and were most commonly used as a dispersed phase in hybrid matrix membranes [198]. The preparation of hybrid matrix membranes by using MIL-101(Cr) or MIL-101(Cr)-based materials is another hotspot of research. Rajati et al. [199] combined polyvinylidene fluoride (PVDF) and MIL-101(Cr) to prepare Matrimid/PVDF/MIL-101(Cr) membranes. This hybrid matrix membrane had excellent permeability and CO_2_/CH_4_ selectivity. Compared with the original matrimid membrane, the hybrid membrane had a 102% increase in CO_2_ permeability and a 77% increase in selectivity. This can be attributed to the high polarization of the MIL-101(Cr) structure and the adsorption of CO_2_ by MIL-101(Cr), which enhanced the solubility of CO_2_ molecules in the membrane and thus improved the permeability of the hybrid matrix membrane. Subsequently, Rajati and coworkers prepared mixed matrix membranes containing ionic liquids and NH_2_-MIL-101(Cr) [23]. Compared with the original matrix membrane, the prepared hybrid membrane possessed a 162% increase in CO_2_ permeability and a 224% increase in selectivity.

## 4. Conclusions

In summary, the structural properties of MIL-101(Cr) were significantly influenced by the synthetic method. At present, the hydrothermal synthesis method is the most commonly used method, which could produce MIL-101(Cr) with high crystallinity and excellent properties. Many scientists used other acidic additives (acetic acid, hydrochloric acid, etc.) instead of hydrofluoric acid to participate in the reaction and successfully synthesized MIL-101(Cr) with excellent properties and high specific surface area and porosity. Other methods, including the microwave-assisted method, the solvothermal method, and the template method, were also fully discussed. Microwave-assisted, high-temperature, and high-pressure conditions could accelerate the synthesis of MIL-101(Cr). MIL-101(Cr), with excellent performance and hierarchical pore structure, could be synthesized by using CTAB and other substances as template agents.

MIL-101(Cr) had high porosity and unsaturated metal sites in its structure, which had excellent adsorption properties for gases, dye solutions, and volatile compounds. Especially, MIL-101(Cr) possessed excellent adsorption capacity for CO_2_ and H_2_. According to the literature, MIL-101(Cr) amine functionalization, carbon doping, and metal doping could greatly improve the adsorption capability of CO_2_ and increase its CO_2_/N_2_ selectivity. MIL-101(Cr) also exhibited great potential application in wastewater treatment due to the efficient removal of organic dyes, drug residues, and heavy metal ions, etc., from aqueous solutions.

The presence of removable water molecules in the structure of MIL-101(Cr) provided potential unsaturated metal sites, which can be used as catalytic sites in various reactions. Additionally, grafting functional groups, combined with metal nanoparticles, metal oxides, or other guests, were common strategies to enhance the catalytic activity of MIL-101(Cr). Furthermore, MIL-101(Cr) could be used as a substrate for drug delivery, proton conduction, and hybrid matrix membranes. However, MIL-101(Cr) is a microporous MOF whose maximum capture window is only ~16 Å. The small pore size is not conducive to the rapid diffusion and transport of molecules, which affects the adsorption and catalytic rates of MIL-101(Cr) and greatly hinders the practical application of MIL-101(Cr) in adsorption and catalysis. Therefore, expanding the pore size of MIL-101(Cr) is the most direct way to improve the performance of MIL-101(Cr), which is also a hot topic of research today. Further studies on the functionalization of MIL-101(Cr) by various functional groups and the combination of MIL-101(Cr) with the guest compound nanomaterials are beneficial for the preparation of multifunctional hybrid materials.

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
