# Peer review of "Advances in Metal-Organic Frameworks MIL-101(Cr)"

_ijms, 2022, doi:10.3390/ijms23169396_

Round 1
Reviewer 1 Report
This is an interesting article where the authors have in detail discussed about the synthesis strategies of Advances in Metal-Organic Frameworks MIL-101(Cr) based nanocomposites. The review is interesting, very timely, and provides huge information, drawn from a significant number of papers and other literature sources
1. Please well summarize the advantages and disadvantages of Metal-Organic Frameworks MIL-101 nanocomposites for adsorption, catalysis, sensing, and Electronic Device applications
2. Please compare the differences between the current Metal-Organic Frameworks MIL-101 nanocomposites-based and other types or emerging types of two-dimensional materials. https://doi.org/10.1016/j.bios.2022.114511
3. Please comment on the potential of Emerging Metal-Organic Frameworks MIL-101 for clinic or potential applications. Whether it can be eventually commercially used for clinics?
Author Response
Dear Reviewers,
Thank you for the refereeing of the above manuscript, for your useful comments, suggestions and the structure of our manuscript. We have modified the manuscript accordingly (which were highlighted in red color), and the detailed corrections are listed below point by point.
We thank the referees for their time in assessing the manuscript.
We hope that our revision has further improved the quality of the work.
Sincerely,
Tian Zhao
- Please well summarize the advantages and disadvantages of Metal-Organic Frameworks MIL-101 nanocomposites for adsorption, catalysis, sensing, and Electronic Device applications
Response: Thank you for your suggestion. We added the relevant content in the manuscript “MIL-101(Cr) had high porosity and unsaturated metal sites in its structure, which had excellent adsorption properties for gases, dye solutions, and volatile compounds. MIL-101(Cr) possessed excellent adsorption capacity for CO2 and H2”. “MIL-101(Cr) also exhibited great potential application in wastewater treatment, due to the efficient removal of organic dyes, drug residues and heavy metal ions, etc. from aqueous solutions”. “The small pore size is not conducive to the rapid diffusion and transport of molecules, which affects the adsorption and catalytic rates of MIL-101(Cr) and greatly hinders the practical application of MIL-101(Cr) in adsorption and catalysis.”
- Please compare the differences between the current Metal-Organic Frameworks MIL-101 nanocomposites-based and other types or emerging types of two-dimensional materials.
Response: Thank you for your suggestion. And we have modified it in the manuscript “Zhang et al. [193] prepared an immunosensor based on nanoparticle-loaded MIL-101(Cr) for the detection of microcystin lr in water. The sensor had good stability and practicality, and had ultra-high recovery for the detection of microcystin lr in water bod-ies. The MIL-101(Cr) sensor had a very good recovery with a detection recovery of 102%, which was higher than that of the nanobiosensor prepared from NiO-rGO\/MXene com-plex (89-101%) [194].”.
- Please comment on the potential of Emerging Metal-Organic Frameworks MIL-101 for clinic or potential applications. Whether it can be eventually commercially used for clinics?
Response: Thank you for your suggestion. In fact, as far as we know, the current research on MIL-101(Cr) in medicine is mainly focused on drug delivery system, MIL-101(Cr) contains Cr element, which is harmful to human cells, therefore, the clinical application of MIL-101(Cr) still meet huge challenge. We added the related content in the manuscript: “ Although MIL-101(Cr) showed good potential for drug delivery application, however, as far as we know, it cannot be eventually commercially used for clinics yet, due to the presence of Cr in the framework.”

Reviewer 2 Report
The manuscript entitled " Advances in Metal-Organic Frameworks MIL-101(Cr)" reviewed the synthesis, application adsorption, and catalysis.
Generally, the manuscript is well organized. However, I do have a few concerns that need to be addressed:
1. The review is focused on MIL-101(Cr). Please revise Figure 1 as MOFs are a too wide topic that the review cannot cover; the statistics of scientific articles related to MIL-101(Cr) need to be shown in Figure 1.
2. Please revise the logic structure of “Chapter 2 Synthesis”. Chapter “2.3. Microwave-assisted method” and “2.4 Template method” can not be categorized at the same level as hydrothermal method and solvothermal method.
3. Please revise the conclusion. A conclusion should be comprehensive. The authors need to summarize the cutting-edge research work and propose research directions for MIL-101(Cr).
4. Other comments
In Figure 2, the red oxygen atom can hardly be seen.
Table 1, some abbreviations need to be corrected, e.g. HCL, CH3COOk…
The inset graph of Figure 4 is vague.
Author Response
Dear Reviewers,
Thank you for the refereeing of the above manuscript, for your useful comments, suggestions and the structure of our manuscript. We have modified the manuscript accordingly (which were highlighted in red color), and the detailed corrections are listed below point by point.
We thank the referees for their time in assessing the manuscript.
We hope that our revision has further improved the quality of the work.
Sincerely,
Tian Zhao
On the technical level:
- The review is focused on MIL-101(Cr). Please revise Figure 1 as MOFs are a too wide topic that the review cannot cover; the statistics of scientific articles related to MIL-101(Cr) need to be shown in Figure 1.
Response: Thank you very much for your suggestion, we have added a graph with statistics about scientific articles related to MIL-101(Cr).
Figure 1. The statistics of scientific articles on 'MIL-101(Cr)' since 1998 (data from ScienceDirect).
- Please revise the logic structure of “Chapter 2 Synthesis”. Chapter “2.3. Microwave-assisted method” and “2.4 Template method” can not be categorized at the same level as hydrothermal method and solvothermal method.
Response: Thanks for the suggestion and we have revised it in the manuscript “2.1. Hydrothermal synthesis method. 2.1.1. Traditional hydrothermal method. 2.1.2. Microwave-assisted hydrothermal method. 2.1.3. Template hydrothermal method”.
- Please revise the conclusion. A conclusion should be comprehensive. The authors need to summarize the cutting-edge research work and propose research directions for MIL-101(Cr).
Response: Thank you very much for your suggestion, we have updated the conclusion. And the new statement was “In summary, the structural properties of MIL-101(Cr) were significantly influenced by the synthetic method. At present, the hydrothermal synthesis method was the most commonly used method, which could produce MIL-101(Cr) with high crystallinity and excellent properties. Many scientists used other acidic additives (acetic acid, hydrochloric acid, etc.) instead of hydrofluoric acid to participate in the reaction and successfully synthesized MIL-101(Cr) with high specific surface area and porosity. Other methods, including microwave-assisted method, solvothermal method and template method, were also fully discussed. Microwave-assisted, high-temperature and high-pressure conditions could accelerate the synthesis of MIL-101(Cr). MIL-101(Cr) with excellent performance and hierarchical pore structure could be synthesized by using CTAB and other substances as template agents.
MIL-101(Cr) had high porosity and unsaturated metal sites in its structure, which had excellent adsorption properties for gases, dye solutions, and volatile compounds. Especially, MIL-101(Cr) possessed excellent adsorption capacity for CO2 and H2.
According to the literatures, MIL-101(Cr) amine functionalization, carbon doping and metal doping could greatly improve the adsorption capability of CO2 and increase its CO2/N2 selectivity. MIL-101(Cr) also exhibited great potential application in wastewater treatment, due to the efficient removal of organic dyes, drug residues and heavy metal ions, etc. from aqueous solutions.
The presence of removable water molecules in the structure of MIL-101(Cr) provided potential unsaturated metal sites, which can be used as catalytic sites in various reactions. Additionally, grafting functional groups, combining with metal nanoparticles, metal oxides or other guests, were common strategies to enhance the catalytic activity of MIL-101(Cr). Also, MIL-101(Cr) could be used as a substrate for drug delivery, proton conduction, and hybrid matrix membranes. However, MIL-101(Cr) is a typical mesoporous MOF with a maximum pore size of 34 Å, but its maximum capture window is only ~16 Å. The small pore size is not conducive to the rapid diffusion and transport of molecules, which affects the adsorption and catalytic rates of MIL-101(Cr) and greatly hinders the practical application of MIL-101(Cr) in adsorption and catalysis. Therefore, expanding the pore size of MIL-101(Cr) is the most direct way to improve the performance of MIL-101(Cr), which is also a hot topic of research today. Further studies on the functionalization of MIL-101(Cr) by various functional groups and the combination of MIL-101(Cr) with the guest com-pound nanomaterials are beneficial for the preparation of multifunctional hybrid materials”.
- In Figure 2, the red oxygen atom can hardly be seen.
Response: We apologize for the image and we have replaced it in the manuscript.
Figure 2. (a) Framework structure of MIL-101. (b) small cage with pentagonal windows and large cage with pentagonal and hexagonal windows with pore diameters (yellow spheres). The yellow spheres in the mesoporous cages with diameter of 29 or 34 Å, respectively, take into account the van-der-Waals radii of the framework walls (water-guest molecules are not shown).
- Table 1, some abbreviations need to be corrected, e.g. HCL, CH3COOk…
Response: We are very sorry for our incorrect writing and it has been corrected in the text.
- The inset graph of Figure 4 is vague.
Response: We apologize for the vague image and we have replaced it in the manuscript.
Figure 4. Synthesis process of MIL-101(Cr) (left) (the inset shows the adsorption capacity curve of the sample for toluene) and SEM images of MIL-101(Cr) (right); (a)MIL-101(Cr) as made; (b)TMAOH-1@MIL-101(Cr); (c)TMAOH-2@MIL-101(Cr); (d)TMAOH-3@MIL-101(Cr) [48]. Re-printed from ref. 48. Copyright © 2022, with permission from Elsevier.

Reviewer 3 Report
In this work, the author discussed most of the examples based on MIL-101 in different field of application. The review is written very well and is of potential for publication. I have some small comments and I wish the author to implement them in the modified version.
1. Author should differentiate between MIL-101 (Cr) vs MIL-101 (Fe) in the introduction.
2. The difference between MIL-family compounds should be discussed briefly.
3. In the synthesis table, the author should include the stability of the MOFs and their Pore size under each synthetic conditions. If some article discussed the particle size, the author should write in the discussion. At the en of the synthesis section, the author should provide a discussion comparing all the synthesis method known so far. The author should provide more information on the best method among them.
4. In Figure 8, Uio-66 should be replaced by UiO-66
5. In all the applications discussed in this review. The author should provide the best performance material for a certain application. For ex. which MOF has best CO2 capacity and compare with all the MIL-101 based materials.
6. At the end, the author should provide future prospective of MIL-101 and discuss what should be done and how MIL-101 can be modified to better materials with better performance.
7. Scale of synthesis of MIL-101 should be discussed.
Thank you
Author Response
Dear Reviewers,
Thank you for the refereeing of the above manuscript, for your useful comments, suggestions and the structure of our manuscript. We have modified the manuscript accordingly (which were highlighted in red color), and the detailed corrections are listed below point by point.
We thank the referees for their time in assessing the manuscript.
We hope that our revision has further improved the quality of the work.
Sincerely,
Tian Zhao
- Author should differentiate between MIL-101 (Cr) vs MIL-101 (Fe) in the introduction.
Response: Thank you very much for your advice. We have revised and rewritten the section, and the new statement was “The MIL-101 series MOFs all have similar zeolite topology but differ in surface morphology, density, and pore size. For example, MIL-101(Fe) and MIL-101(Cr) have the same topology and framework structure, and both of them are well studied. MIL-101(Fe) is composed of Fe(III) octahedral chains as secondary building units (SBU) and 1,4-benzenedicarboxylic acid [17]. MIL-101(Fe) has good catalytic properties, and under certain conditions, part of the Fe3+ in MIL-101(Fe) will be converted to Fe2+, which can play a good activation role in catalytic applications.
MIL-101(Cr) is one of the most representative material of the MIL series and one of the most investigated MOFs today. Scientific research reports on the topology and potential applications of MIL-101(Cr) materials have continued to grow over the last 30 years (as shown in Figure 1). MIL-101(Cr) is formed by coordination of Cr3O ionic cluster with terephthalic acid (H2BDC), with the formula [Cr3(O)X(BDC)3(H2O)2] Microwave Irradiation (where BDC is terephthalic acid and X is OH- or F-) [18], and its structure is similar to the MTN zeolite topology, as shown in Figure 2a. MIL-101(Cr) possesses two different sizes of mesoporous cage cavities with diameters of 29 Å and 34 Å (Figure 2b), and the pore windows can reach 16 Å in diameter, with a Brunauer-Emmet-Teller (BET) specific surface area of 4100 m2 g-1. MIL-101(Cr) has crystalline water molecules at the end of its molecular structure, which can be removed under high temperature or vacuum conditions, making MIL -101(Cr) have unsaturated metal sites (i.e., possessing potential Lewis acidic sites) [19]. MIL-101(Cr) has very high porosity, good physicochemical properties and chemical stability, thus, it is widely used in electrocatalysis [20], photocatalysis [21], pollutant adsorption [22], mixed matrix membranes [23], detection [24], drug transport [25] and other important fields.”
- The dfference between MIL-family compounds should be discussed briefly.
Response: Thank you for your professional suggestion. We have added descriptions of relevant content and the new statement was “Materials of Institute Lavoisier Frameworks (MIL) materials are one of the most studied materials for MOFs. Materials of Institute Lavoisier Frameworks (MIL) materials are one of the most studied materials for MOFs. M(III) terephthalates (M = Cr, Fe, Al, V, Mn, In in decreasing order of importance as well as some others) together with terephthalate derivatives and elongated terephthalate analogues form a particularly important sub-class of MOFs. The four best known porous M(III) terephthalates (and terephthalate analogues) are MIL-47/MIL-53, MIL-88, MIL-100 and MIL-101. Most of the MIL series materials use Cr3+, Fe3+, Al3+ as metal ion clusters with terephthalate derivatives and terephthalate analogs as organic ligands to ligand [5]. The MIL-101 series MOFs all have similar zeolite topology but differ in surface morphology, density, and pore size.”
- In the synthesis table, the author should include the stability of the MOFs and their Pore size under each synthetic conditions. If some article discussed the particle size, the author should write in the discussion. At the en of the synthesis section, the author should provide a discussion comparing all the synthesis method known so far. The author should provide more information on the best method among them.
Response: Thank you for your suggestion, we have added particle size data for MIL-101(Cr) under different synthesis conditions. However, the stability of the MOFs and their pore size under each synthetic conditions are the same in the literatures. Thus, it’s not necessary to list them in the table.
We have added the discussion comparing all the synthesis method known so far in the end of the synthesis section. “At present, hydrothermal method was still the most commonly used strategy for MIL-101(Cr) synthesis, which possesses high specific surface area and good crystallinity. And numerous scientists tried to use other additives to replace the original reported HF during MIL-101(Cr) synthesis. For example, the use of HNO3 could rise the yield over 80 %, the use of acetic acid could achieve nano-sized product, etc. Microwave-assistant method would largely accelerate the reaction process and save the time, while template method could provide special structural MIL-101(Cr) crystals and solvothermal method could decrease the reaction temperature as low as 140 oC. Thus, the researchers could choose the appropriate synthesis method according to their needs and conditions.”
- In Figure 8, Uio-66 should be replaced by UiO-66.
Response: Thank you very much for your suggestion. We have changed the Uio-66 to UiO-66 in Figure 8.
- In all the applications discussed in this review. The author should provide the best performance material for a certain application. For ex. which MOF has best CO2 capaity and compare with all the MIL-101-based materials.
Response: Thanks for the suggestions. I have added the relevant content at the conclusion, and the new statement was “Yang et al. [112] used sulfonyl modification of MIL-101 to produce MIL-101-SO3H, which was found to have excellent adsorption properties for organic dyes. The experimental adsorption capacities of MIL-101-SO3H for methyl orange, Congo red and acid chromium blue K were 688.9, 2592.7 and 213.2 mg g-1, which were 69.6%, 89.6% and 51.5% higher than those of unmodified MIL-101, respectively. As it can be seen in Table 3, the adsorbent had the best dye adsorption performance in MIL-101(Cr) and derivatives. The uncoordinated -SO3H group increased the electrostatic attraction and hydrogen bonding between MIL-101-SO3H adsorbent and linear anionic dyes, thus increasing the adsorp-tion capacity of the linear anionic dyes.”
- At the end, the author should provide future prospective of MIL-101 and discuss what should be done and how MIL-101 can be modified to better materials with better performance.
Response: Thank you for your suggestions. I have added the relevant content at the conclusion, and the new statement was “Therefore, expanding the pore size of MIL-101(Cr) is the most direct way to improve the performance of MIL-101(Cr), which is also a hot topic of research today. Further studies on the functionalization of MIL-101(Cr) by various functional groups and the combination of MIL-101(Cr) with the guest compound nanomaterials are beneficial for the preparation of multifunctional hybrid materials.”
- Scale of synthesis of MIL-101 should be discussed.
Response: Thank you very much for your advice. As far as we know, the current synthesis of MIL-101(Cr) and its derivatives were small-scale laboratory synthesis (5 ~ 20 mL), and large-scale synthesis as well as industrial production have not been reported yet.
